# The Trajectory of Nutritional Status and Physical Activity before and after Transcatheter Aortic Valve Implantation

**DOI:** 10.3390/nu14235137

**Published:** 2022-12-02

**Authors:** Dennis van Erck, Christine D. Dolman, Wilma J. M. Scholte op Reimer, José P. Henriques, Peter J. M. Weijs, Ronak Delewi, Josje D. Schoufour

**Affiliations:** 1Cardiology, Amsterdam UMC, University of Amsterdam, Meibergdreef 9, 1105 AZ Amsterdam, The Netherlands; 2Cardiothoracic Surgery, Amsterdam UMC, University of Amsterdam, Meibergdreef 9, 1105 AZ Amsterdam, The Netherlands; 3Research Group Chronic Diseases, HU University of Applied Sciences Utrecht, Padualaan 99, 3584 CH Utrecht, The Netherlands; 4Faculty of Sports and Nutrition, Amsterdam University of Applied Sciences, Dokter Meurerlaan 8, 1067 SM Amsterdam, The Netherlands; 5Faculty of Health, Amsterdam University of Applied Sciences, Dokter Meurerlaan 8, 1067 SM Amsterdam, The Netherlands

**Keywords:** nutritional status, physical activity, medical procedure, TAVI, older patients

## Abstract

It is suggested that older patients waiting for an elective surgical procedure have a poor nutritional status and low physical activity level. It is unknown if this hypothesis is true and if these conditions improve after a medical procedure. We aimed to determine the trajectory of both conditions before and after transcatheter aortic valve implantation (TAVI). Included patients (n = 112, age 81 ± 5 years, 58% male) received three home visits (preprocedural, one and six months postprocedural). Nutritional status was determined with the mini nutritional assessment—short form (MNA-SF) and physical activity using an ankle-worn monitor (Stepwatch). The median MNA-SF score was 13 (11–14), and 27% of the patients were at risk of malnutrition before the procedure. Physical activity was 6273 ± 3007 steps/day, and 69% of the patients did not meet the physical activity guidelines (>7100 steps/day). We observed that nutritional status and physical activity did not significantly change after the procedure (β 0.02 [95% CI −0.03, 0.07] points/months on the MNA-SF and β 16 [95% CI −47, 79] steps/month, respectively). To conclude, many preprocedural TAVI patients should improve their nutritional status or activity level. Both conditions do not improve naturally after a cardiac procedure.

## 1. Introduction

Severe loss of skeletal muscle strength, muscle mass, and physical performance, known as sarcopenia, is a major problem in the older population [1]. Sarcopenia contributes to impairments in physical function, immobility, and dependence and thereby negatively influences (healthy) life years [2,3]. Methods to prevent sarcopenia in older adults are therefore of great importance. Decline in muscle strength, muscle mass, and physical performance is influenced by many factors (e.g., age, environment, or genetic factors). Physical inactivity and poor diet are the most important modifiable factors contributing to the onset and severeness of sarcopenia [1]. Consequently, events leading to a steep decline in physical activity and/or poor dietary intake, such as a medical event or hospital admission, can accelerate the process of sarcopenia [1,4,5].

Patients on the waiting list for a medical procedure are particularly at risk of a steep decline in physical inactivity and poor dietary intake [6,7,8]. Consequently, rehabilitation and prehabilitation interventions focusing on a combination of dietary intake and physical activity have shown positive effects on the reduction of complications and improvement of physical functioning and quality of life [9,10]. Patients with higher vulnerability and higher age are at an even higher risk for poor dietary intake and physical inactivity [11]. Therefore, it is suggested that especially these older vulnerable patients could benefit from interventions improving nutritional status and physical activity [11,12]. However, in these older patients, rehabilitation or prehabilitation is generally not part of usual care, and vulnerable patients are often even excluded for research on pre- and rehabilitation due to practical barriers and uncertainty about the effectiveness and motivation in this population [13,14].

A typically vulnerable and older patient population, with an average age above 80 years old, are patients with severe aortic stenosis undergoing an elective transcatheter aortic valve implantation (TAVI) [15]. Severe aortic stenosis is the most common heart valve disease, with increased prevalence at older age. Prevalence is approximately 3.4% among adults above the age of 75 years [16]. Reported symptoms of severe aortic stenosis are dyspnea, fatigue, and syncope [17]. Despite low invasiveness of the transcatheter intervention, it is shown that approximately one in three patients experience physical decline or are deceased within six months after the procedure [18]. Poor nutritional status and physical inactivity during the preprocedural period are shown as predominant risk factors for negative outcomes [19,20,21]. It is suggested that in this patient population, prehabilitation could be effective in improving outcomes [11,12]. However, traditional center-based approaches and currently available home-based interventions are not immediately applicable to this older population because of barriers such as transportation or complexity and intensity of exercises. Therefore, new interventions should be developed specifically for this patient group [11,14].

Before feasible and effective interventions can be developed, more knowledge is required on objectively measured physical activity and the trajectory of nutritional status and physical activity after a TAVI procedure [22]. This knowledge can provide useful insight on the timing of an intervention (e.g., only before or also after the procedure), the focus of the intervention, and the selection of patients that could benefit from the intervention [11,12,23]. The aim of this longitudinal study was to determine the course of nutritional status and physical activity from preprocedural up to six months after the procedure. The second aim was to explore patient characteristics associated with nutritional status and physical activity and with changes in nutritional status and physical activity.

## 2. Materials and Methods

### 2.1. Study Design and Participants

This study is a prospective observational cohort. All consecutive patients planned for an elective TAVI in our high-volume tertiary cardiac care and TAVI center were asked to participate. Inclusion took place from January 2020 until September 2021. Exclusion criteria were the inability to speak Dutch, more than two hours travel distance from home to the hospital, and no time or ability to perform preprocedural measurements. At enrollment, all included patients were asked to sign the informed consent form. The study was performed in accordance with the Declaration of Helsinki and has been approved by the local medical ethics committee (W19_450).

### 2.2. Data Collection

Data were collected during three home visits by a researcher or trained research assistant. Home visits took place approximately six weeks before the procedure, as soon as possible after patients visited the hospital for their preprocedural screening, 30 days after the procedure, and six months after the procedure. During these home visits, a comprehensive battery of tests was performed to assess the physical and psychological status of the patient.

#### 2.2.1. Nutritional Status and Physical Activity

Nutritional status was measured with the mini nutritional assessment—short form (MNA-SF), which is a valid tool for screening on malnutrition in older adults [24]. A score on the MNA-SF of <12 points indicates a risk of malnutrition, and a score of <7 indicates malnutrition. Physical activity was objectively measured as time-stamped steps with the Stepwatch 4 (Modus Health LLC, Edmonds, WA, USA). The Stepwatch is a validated tool for measuring physical activity in older adults with normal, slow, or irregular gait speed [25]. The Stepwatch was worn by patients during every awake hour for seven consecutive days. Data were visually checked by one researcher (D.E.) and was only included if the device was worn for a minimum of ten hours per day [26]. Patients with less than three complete days were excluded [26]. Physical activity measures included the number of steps per day and the peak 30 min cadence (average steps/min recorded for the highest 30 non-consecutive minutes in a day) [27]. The number of steps per day is a good marker of overall physical activity, while the 30 min peak cadence represents the intensity of the activity [27]. In addition, the number of patients with low activity level and low intensity were reported. A low activity level was defined as <7100 steps per day because 7100 steps are in accordance with the international activity guidelines [28]. For the peak 30 min cadence, a threshold of 70 steps/min was used, which is associated with better health outcomes [27,29].

#### 2.2.2. Measures of Patient Characteristics

Depression was measured with the geriatric depression scale (GDS-15), with a score range between 0 and 15 and a score of 6 or higher indicating possible depression [30]. Cognition was measured with the mini-mental state examination (MMSE), which is a test of 11 questions with a maximum score of 30 points and a score < 22 points indicating lower cognition [31]. Anxiety was measured with the cardiac anxiety questionnaire (CAQ), which is a useful screening tool to determine heart-focused anxiety [32]. The CAQ is comprised of 18 questions on a 5-point Likert scale, with a higher score indicating higher cardiac anxiety. Lower body functioning was measured with the short physical performance battery (SPPB), which is a combination of three tests, namely standing balance, three-meter gait speed, and five-time chair stand test. For every test, 4 points can be scored, adding up to a total score of 12. A score of 8 points or lower indicates a lower performance [33]. Physical capacity was measured with the two-minute step test, in which patients had to lift their knee to a height midway between the patella and iliac crest, as much as possible within two minutes [34]. More steps during the two minutes indicate a higher physical capacity. Lower body strength was measured with the five-time chair stand test, in which patients had to stand up five times from a chair as fast as possible, with less time indicating more strength [35]. Muscle strength was measured with a handheld dynamometer and was determined as the highest value of three attempts. A handgrip strength < 27 kg for males and <16 kg for females indicates low strength. Furthermore, patients were asked to rate their fatigue, fear of falling, and pain on a ten-point numeric rating scale (NRS). Additionally, the presence of conditions as dyspnea, fatigue, and syncope were determined. Other collected patient characteristics were age, sex, body mass index (BMI), living arrangement, surgical risk score (Euroscore-II), New York Heart Association score (NYHA), chronic obstructive pulmonary disease (COPD), history of coronary heart disease, peripheral artery disease, left ventricular ejection fraction (LVEF), and access site of the procedure (transfemoral or transaortic).

### 2.3. Statistical Analysis

Results are reported as mean and standard deviation or median and interquartile range depending on distribution or as amount and percentage. All analyses were performed in R version 4.2.0 (The R Foundation, Vienna, Austria).

To determine the changes in nutritional status and physical activity, descriptive statistics and a linear mixed model were performed. In the linear mixed model, time was set as a fixed factor and subject as random intercept. Time as a random slope was inserted when indicated. Interaction effects of sex with changes in nutritional status and physical activity over time were used to check if stratification was needed. Thereafter, the model was adjusted for age and sex, and the following potential confounders were considered: BMI, COPD, cognition (MMSE), depression (GDS), cardiac anxiety (CAQ), physical performance (SPPB), history of coronary heart disease, peripheral artery disease, LVEF, NYHA, fatigue, living arrangement, Euroscore- II, access site of the procedure, fear of falling, and pain. When the beta of interest changed more than 10%, the potential confounder was included in the fully adjusted model. For our second aim, to explore the association of patient characteristics with nutritional status and physical activity, a second linear mixed model was performed with the subject as a random intercept. Baseline geriatric conditions and other patient characteristics were evaluated in this analysis. Based on the univariate analysis, variables with a *p*-value < 0.10 were entered in the multivariable model. Thereafter, a backward selection was performed to determine characteristics independently associated with nutritional status and physical activity. Baseline characteristics associated with changes in nutritional status (MNA-SF) and physical activity (steps/day) were calculated using a multivariate linear model. To determine the effect of missing data, sensitivity analyses were performed with complete case and worst-case scenario.

## 3. Results

### 3.1. Patient Characteristics

From 198 eligible patients, 112 patients were included. The mean age of the included patients was 81 ± 5 years, and 58% (n = 66) were male (Table 1). Measurements before the procedure took place directly after acceptance for the TAVI and was a median of 49 (19–80) days before the procedure. At six months, seventeen patients were lost to follow-up, five patients were deceased, nine patients indicated that they were not fit enough to further participate, and three patients were unable to be contacted.

### 3.2. Nutritional Status and Physical Activity Level before and after the TAVI

Preprocedurally, 27% of the patients were at risk of malnutrition, and no significant change in nutritional status was observed until six months after the procedure (β 0.02 [95% CI: −0.03–0.07] points/months, *p* = 0.50, Table 2). The individual components of the MNA-SF also showed no significant change (see Appendix A). Although on a group level no differences were seen in the MNA-SF score, there were many individual differences in changes with a range from −4 to +5 points (Figure 1). In total, 69% of the preprocedural TAVI patients did not meet the guidelines for physical activity (>7100 steps/day). Patients took on average 6273 ± 3007 steps per day before the procedure, and activity levels did not significantly change after the procedure (β 16 [95% CI: −47, 79] steps/month, *p* = 0.62). For physical activity, there were also many individual differences between patients regarding the changes in physical activity before and after the procedure. The patient with the least change showed a difference of 8 steps per day, and the patient with the highest change showed a difference of 6429 steps per day (Figure 2). The peak 30 min cadence was on average 66 ± 20 steps/min before the procedure, and 61% of the patients did not meet the threshold of intensity associated with better long-term health. The peak 30 min cadence showed no change after the procedure (β 0.02 [95% CI: −0.41–0.45] steps/month, *p* = 0.94). The sensitivity analyses on complete cases and the worst-case scenario showed no remarkable differences from the original analysis (Appendix A).

### 3.3. Patient Characteristics Associated with Nutritional Status and Physical Activity Level

The linear mixed model on all timepoints showed four patient characteristics associated with nutritional status and four with physical activity. Lower BMI, lower cognition, lower handgrip strength, and more pain were associated with a worse nutritional status (Table 3); higher BMI, COPD, fear of falling, and worse physical performance were associated with a lower physical activity level.

### 3.4. Baseline Characteristics Associated with Change in Nutritional Status and Physical Activity Level

In the multivariate linear model, higher baseline depression and fatigue were significantly associated with a decline in nutritional status from preprocedural until six months (Table 4). Higher handgrip strength and higher baseline depression as assessed with the GDS predicted an increase in physical activity.

## 4. Discussion

We observed that one in four patients have a poor nutritional status before the procedure and more than two-thirds of the patients do not meet the international guidelines of physical activity. Neither nutritional status nor physical activity changed after the procedure on group level. Large differences were observed between individuals. Higher levels of fatigue and depression were predictive for a decline in nutritional status, while higher depression and higher handgrip strength were associated with an increase in physical activity.

Our observation that 27% of the preprocedural TAVI patients is at risk of malnutrition is in line with studies in community-living healthy older adults of the same age [36]. Factors associated with lower nutritional status were lower BMI, lower cognition, lower handgrip strength, and more pain. The association with BMI and cognition seems self-evident, as these factors are elements of the MNA-SF screening tool. In addition, lower handgrip strength is an expected predictor of nutritional status because handgrip strength is strongly and frequently associated with malnutrition [37]. Pain has shown to be closely related to a decline in appetite, which could explain the association with poor nutritional status in our study [38]. In the work-up before TAVI, screening for malnutrition risk is important for the heart team to make informed treatment decisions about the acceptance of a patient for the TAVI procedure. When malnutrition risk is detected, involvement of a dietitian before the procedure should be considered, which is currently not described in the guidelines for TAVI procedure [15,39].

For physical activity, we observed on average 6500 steps per day and a peak 30 min cadence of 66 min/month. Both the number of steps and intensity are similar to community living older adults, with an average number of steps between 6000 and 7000 steps per day and peak 30 min cadence around the 70 [27,28,40]. This amount of activity is higher than expected, as most patients undergoing TAVI experience symptoms such as dyspnea, fatigue, and syncope. Although the physical activity level is higher than expected, similar as the community living older adults, the large majority of our study population (69%) did not meet the physical activity guidelines, and 61% did not meet the guidelines for intensity [27]. Although patients are equally as active as their community-living counterparts, the majority of individuals require more physical activity. There is a wide range in number of steps within individual older adults, which we also observed in the preprocedural TAVI population. The number of steps per day in our study ranged from 1500 in the least active patient to almost 18,000 steps per day in the most active individual. This difference reflects the diversity and capability in lifestyle behavior of the older population [28]. Patients with COPD, higher BMI, lower physical performance, and higher fear of falling were found to be less active. As physical activity measurements are time-consuming, screening on these conditions can detect patients at risk of a low activity level.

We observed that neither dietary intake nor physical activity showed significant changes after the TAVI procedure. It is hypothesized that improved physical capacity after TAVI would lead to an increase in physical activity and indirectly to a better nutritional status by improved appetite and lower inflammation [20,41]. However, earlier studies already suggested that this hypothesis does not hold for TAVI patients because an increase in physical capacity does not simply lead to altered behavior [20]. Barriers for many patients to eat healthy or be more active is not only influenced by the physical capacity but also by environmental opportunity, psychological capability, and motivation [42]. Our results confirm that the TAVI procedure is not sufficient to improve the nutritional status and physical activity of patients within six months after the procedure. However, there were large individual differences in changes, as some patients still had major declines or improvements in both conditions. These differences were partly explained by depression score, fatigue, and handgrip strength. Higher depression score and higher self-reported fatigue at baseline were associated with a decline in nutritional status. Both fatigue and depression are strongly associated with reduced appetite and a decline in food intake, leading to weight loss and poor dietary status [43]. For physical activity, our results showed that higher depression and handgrip strength were associated with an increase in physical activity. The association with higher handgrip strength could be explained by the physical condition, which is needed to increase the physical activity level [44]. One explanation for the association between higher baseline depression and increase in physical activity could be the decrease in depression after the TAVI procedure [45]. As physical activity is cross-sectionally associated with depression, this can explain the contradicting association [46,47].

Poor nutritional status and low physical activity level are highly prevalent in patients undergoing TAVI, and this number should be reduced [19]. Multiple studies have shown that malnutrition risk and physical inactivity are the most predominant modifiable risk factors for functional decline or mortality in the first year after TAVI [19,20]. The current study provided valuable insights for the development of nutrition and physical activity interventions for this older patient population. First, our study showed that simply reducing wait time for an elective medical procedure is not sufficient to avoid malnutrition risk and inactivity, as both conditions did not alter from community-living older adults and did not change after the medical procedure. Therefore, rehabilitation before and after hospital admission are needed to increase the activity level. Regarding dietary intake, evidence shows that protein intake in combination with diet quality is important to prevent a decline in muscle mass and physical performance [48,49]. Additionally, in TAVI patients, it is shown that malnutrition is associated with a lower protein intake and lower intake of whole grains, which should be the focus of a nutrition intervention in these patients [8]. During the preprocedural period, there is a contraindication to exercise at moderate or high intensity. However, mild exercise not provoking symptoms may be considered to maintain general health and is often well-tolerated [11,20,50]. As most patients in our study had a physical activity level under the minimally recommended intake, functional exercises at low intensity are warranted to avoid loss of muscle mass and muscle function [1]. After the procedure, exercise intensity can be gradually increased [4,51]. Since patients had similar performance as their healthy counterparts, e-health interventions that were tested in community older adults can possibly also be used in the TAVI population to improve dietary intake and physical activity. In the Netherlands, the VITAMIN trial has shown to increase muscle mass, muscle strength, and physical performance with a blended home-based intervention with progressive functional exercises and dietary counseling [52]. Whether this type of intervention is feasible and effective in the TAVI population is of interest for future studies.

Our study has several limitations. First, we were not able to complete all follow-up data mainly due to mortality or low health status. This could have introduced some survival bias. However, sensitivity analyses with complete case analyses and worst-case scenario analyses did not alter our findings. Second, steps per day are a good and reliable proxy to determine a person’s physical activity level; however, some upper-body physical activities are underestimated by an ankle-worn activity tracker. Lastly, we used the MNA-SF as instrument to determine changes in nutritional status. However, the MNA-SF is mainly developed as a screening tool for malnutrition and not an instrument to diagnose malnutrition. Changes in dietary intake and diagnosed malnutrition in the TAVI population are yet unknown and should be further investigated.

## 5. Conclusions

In total, 27% of older patients waiting for a TAVI procedure were at risk of malnutrition, and 69% did not meet the minimum amount of physical activity according to international guidelines. Nutritional status and physical activity did not significantly change after the TAVI procedure. Future research should focus on the development of pre and post-TAVI interventions for older patients to improve nutritional status and physical activity for better long-term outcomes.

## Figures and Tables

**Figure 1 nutrients-14-05137-f001:**
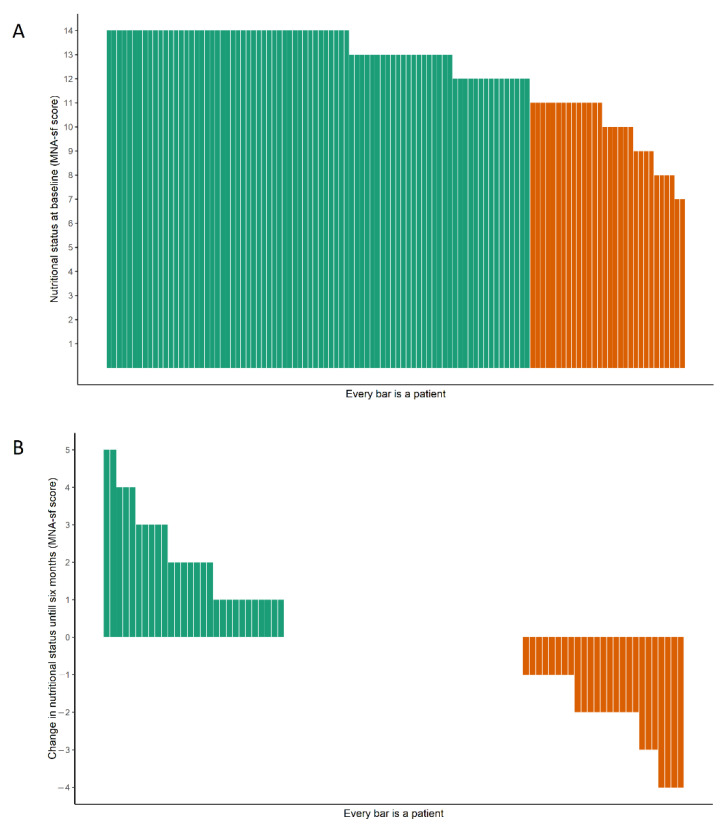
Waterfall plot of (**A**) baseline nutritional status (MNA-SF) and (**B**) change in nutritional status from preprocedural until six months. Green: in (**A**), normal nutritional status; in (**B**), increase in the mini nutritional assessment short form score (MNA-SF). Orange: in (**A**), risk at malnutrition; in (**B**), decrease in the mini nutritional assessment short form score (MNA-SF).

**Figure 2 nutrients-14-05137-f002:**
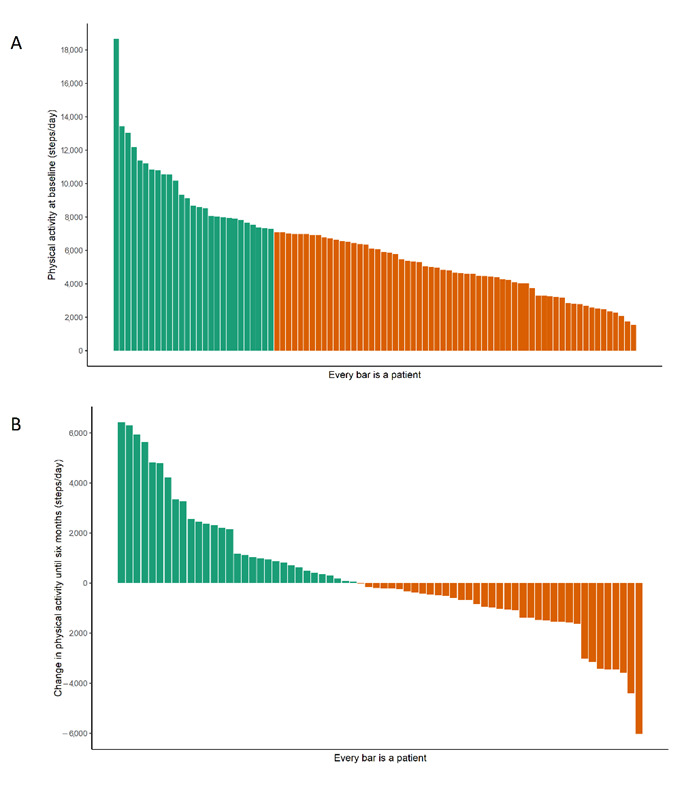
Waterfall plot of (**A**) baseline physical activity and (**B**) change in physical activity from preprocedural until six months. Green: in (**A**), above threshold for recommended minimal activity level (7100 steps/day); in (**B**), increase in activity level. Orange: in (**A**), below threshold for recommended minimal activity level (7100 steps/day); in (**B**), decrease in activity level.

**Table 1 nutrients-14-05137-t001:** Patient characteristics at baseline.

N	112
Demographics Age (years), mean ± SD	81 ± 5
Sex, male, n (%)	66 (58)
BMI, mean ± SD	26.6 ± 4.4
Living with partner (%)	60 (54)
Clinical characteristics Euroscore II, median (IQR)	2.2 (1.6–3.0)
NYHA III/IV, n (%)	53 (47)
COPD, n (%)	13 (11.6)
AVA, mean ± SD	0.76 ± 0.17
LVEF, n (%)	
Good	75 (66.4)
Moderate	34 (30.2)
Poor	2 (1.7)
Transfemoral TAVI, n (%)	102 (91.9)
History of coronary disease, n (%)	32 (29)
History of peripheral arterial disease, n (%)	11 (10)
MNA-SF malnutrition risk, n (%)	30 (27)
Psychosocial factors	
Cardiac anxiety questionnaire, mean ± SD	18 ± 9
Geriatric depression score—15, median (IQR)	3 (2–4)
Mini-mental state score, median (IQR)	28 (27–29)
Dyspnea, n (%)	51 (46)
Fatigue, median (IQR)	5 (2–8)
Syncope, n (%)	34 (30)
Pain, median (IQR)	0 (0–5)
Fear of falling, median (IQR)	2 (0–5)
Physical factors SPPB score, points, mean ± SD	7.8 ± 2.8
Handgrip strength, kg, mean ± SD	
Male	36 ± 10
Female	24 ± 6
TMST, steps, mean ± SD	52 ± 15
Chair stand test, seconds, mean ± SD	15.7 ± 5.2

BMI, body mass index; NYHA, New York Heart Association score; COPD, chronic obstructive pulmonary disease; AVA, aortic valve area; LVEF, left ventricular ejection fraction; SPPB, short physical performance battery; TMST, two-minute step test.

**Table 2 nutrients-14-05137-t002:** Nutritional status and physical activity preprocedural and at 30 days and 6 months after the procedure.

Variable	Preprocedural	30 Days	6 Months	Adjusted Mixed Linear Model of Change per Month		
				β	95% CI	*p*
Nutritional status						
MNA-SF score, median (IQR)	13 (11–14)	13 (12–14)	13 (12–14)	0.02	−0.03, 0.07	0.50
At risk of malnutrition ^1^, %	27	24	22	−0.81	−3.38, 1.75	0.53
Physical activity						
Steps per day, mean ± SD	6273 ± 3007	6708 ± 3113	6866 ± 3233	16	–47, 79	0.62
Low physical activity ^2^, %	69	53	52	−0.07	−0.18, 0.03	0.16
Peak 30 min cadence, mean	66 ± 20	70 ± 22	70 ± 22	0.02	−0.41, 0.45	0.94
Low intensity (<70 steps/min), %	61	51	52	−0.01	−0.11, 0.09	0.81

^1^ Mini nutritional assessment short form (MNA-SF) < 12 points; ^2^ less than 7100 steps/day.

**Table 3 nutrients-14-05137-t003:** Patient characteristics associated with nutritional status and physical activity at all measurement moments.

	Nutritional Status (MNA-SF Score)			Physical Activity (Steps per Day)		
	β	95% CI	*p*	β	95% CI	*p*
Body mass index, per point	0.10	0.04, 0.15	<0.01	−162	−265, −59	<0.01
COPD, yes	-	-	-	−1831	−3318, −345	0.01
Cognition, per point on the MMSE	0.09	0.01, 0.17	0.03	-	-	-
Fear of falling, per point on the NRS (1–10)	-	-	-	−130	−240, −19	0.02
Handgrip strength, per kg	0.04	0.02, 0.07	<0.01	-	-	-
Pain, per point on the NRS (1–10)	−0.11	−0.19, −0.04	<0.01	-	-	-
Physical performance, per point on the SPPB	-	-	-	336	199, 476	<0.01

Other considered variables not associated with nutritional status or physical activity were age, sex, living arrangement, history of peripheral artery disease, history of coronary artery disease, left ventricular ejection fraction, transaortic TAVI, depression (geriatric depression score), cardiac anxiety questionnaire, dyspnea, fatigue (NRS), syncope, physical capacity (two minutes step test), and leg power (chair stand test). COPD, chronic obstructive pulmonary disease; MMSE, mini mental state examination; NRS, numeric rating scale; SPPBs short physical performance battery.

**Table 4 nutrients-14-05137-t004:** Patient characteristics associated with change in nutritional status and physical activity in multivariate analysis.

	Δ Nutritional Status (MNA-SF Score) Preprocedural–6 Month			Δ Physical Activity (Steps per Day) Preprocedural–6 Months		
	β	95% CI	*p*	β	95% CI	*p*
Fatigue, per point on NRS (1–10)	−0.14	−0.26, −0.03	0.01	-	-	-
Depression, per point on GDS	−0.15	−0.26, −0.03	0.01	276	96, 455	<0.01
Handgrip strength, per kg	-	-	-	71	17, 124	0.01

Other considered variables not associated with Δ nutritional status or Δ physical activity were: age, sex, BMI, chronic obstructive pulmonary disease, history of coronary disease, left ventricular ejection fraction, history of peripheral artery disease, cognition (mini mental state examination), fear of falling (NRS), pain (NRS), dyspnea, fatigue (NRS), syncope, cardiac anxiety. NRS, numeric rating scale; GDS, geriatric depression scale; MNA-SF, mini nutritional assessment—short form.

## Data Availability

The data presented in this study are openly available in FigShare at doi.org/10.21942/uva.21347016 (accessed on 19 October 2022).

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
