# Peer review of "The Trajectory of Nutritional Status and Physical Activity before and after Transcatheter Aortic Valve Implantation"

_nutrients, 2022, doi:10.3390/nu14235137_

Round 1

Reviewer 1 Report

The article is made very precisely and great effort was made in making this research.
Paper’s layout is correct, in line with the guidelines for writing research papers. The references are relevant and up-to-date. The number of tables and figures add value to the article, as well as very extensive statistical analysis.
Minor editorial errors in the references are needed. Please check and unify the formatting.
I suggest changing the title because the current version is more like a hypothesis or conclusion.

Author Response

First, we want to thank the reviewer for the nice comments and valuable suggestions. We carefully checked the reference list and made the needed adjustments. We agree that the title of the paper could be improved and we adjusted accordingly towards: ‘The trajectory of nutritional status and physical activity before and after transcatheter aortic valve implantation.’ Additionally, we did extensive language editing to further improve the readability of the paper.  

Reviewer 2 Report

Dear Authors,

Overall, the article is interesting.

My suggestions are the following:

In the introduction, please briefly describe the occurrence and symptoms of severe aortic stenosis in the elderly.
Please indicate the general condition of such patients according to the research.

The mean age in the study group was high, which could be an independent factor.
Depressive disorders and fatigue are related to age, although the average measurement of handgrip strength in the study group is satisfactory.
Regarding the low level of activity as measured by the number of steps, the mean of 6273 ±3007 in this age group is not bad. Take into account the time of the year when the test was performed (3-4 months were still cold).
One should take into account the specificity of bio-psycho-social changes that are taking place and not expect too spectacular results.

The authors are right regarding introducing pre- and rehabilitation in such a group of patients.

Paragraph 2.2.2 Measures of patient characteristics, should be similarly described as 2.2.1.

Geriatric depression score, median [IQR]  (table 1) points to a 5-point version? ( table 1).

Author Response

We want to thank the reviewer for the nice comments and valuable suggestions. Below is a point-by-point reaction to the given comments. Additionally, we did extensive language editing to further improve the readability of the paper. 

In the introduction, please briefly describe the occurrence and symptoms of severe aortic stenosis in the elderly. Please indicate the general condition of such patients according to the research.

Thank you for this important suggestion. Prevalence and symptoms of severe aortic stenosis are added to the introduction. Additionally, we added the prevalence of symptoms in our population to Table 1.

“Severe aortic stenosis the most common heart valve disease, with increased prevalence at older age. Prevalence is approximately 3.4% among adults above the age of 75 years [16]. Reported symptoms of severe aortic stenosis are dyspnea, fatigue and syncope [17].”

The mean age in the study group was high, which could be an independent factor.

Our patient group indeed had a high age above 80 years old. However, this is common for the TAVI population and agrees with other studies in this patient group. We added information to the introduction for clarification.

“A typically vulnerable and older patient population, with an average age above 80 years old, are patients with severe aortic stenosis undergoing and elective transcatheter aortic valve implantation (TAVI) [15].”   

Regarding the low level of activity as measured by the number of steps, the mean of 6273 ±3007 in this age group is not bad. Take into account the time of the year when the test was performed (3-4 months were still cold).

We completely agree that the amount of activity is not bad for this patient group and was higher than expected. We made some changes in the discussion to make this clear.

“Both the number of steps and intensity are similar to community living older adults, which showed an average number of steps between 6000 and 7000 steps per day and peak 30 minute cadence around the 70 [27,28,40]. This amount of activity is higher than expected as most patients undergoing TAVI experience symptoms as dyspnea, fatique and syncope. Although the physical activity level is higher than expected, similar as the community living older adults, the large majority of our study population (69%) did not meet the physical activity guidelines and 61% did not meet the guidelines for intensity [27].”   

The authors are right regarding introducing pre- and rehabilitation in such a group of patients.

Thank you for this comment. We currently work on the design of an RCT to test a pre- and rehabilitation intervention in this population. 

Paragraph 2.2.2 Measures of patient characteristics, should be similarly described as 2.2.1.

We agree and adjusted the complete paragraph as suggested.

“2.2.2 Measures of patient characteristics. 

Depression was measured with the geriatric depression scale (GDS-15), with a score range between 0 and 15 and a score of 6 or higher indicating possible depression [30]. Cognition was measured with the mini-mental state examination (MMSE), which is a test of 11 questions with a maximum score of 30 points and a score <22 points indicating lower cognition [31]. Anxiety was measured with the cardiac anxiety questionnaire (CAQ), which is a useful screening tool to determine heart-focused anxiety [32]. The CAQ is comprised of 18 questions on a 5 point Likert scale with a higher score indicating higher cardiac anxiety. Lower body functioning was measured with the short physical performance battery (SPPB) which is a combination of three tests, which are standing balance, three meter gait speed and five time chair stand test. For every test 4 points can be scored adding up to a total score of 12. A score of 8 points or lower indicates a lower performance [33]. Physical capacity was measured with the two-minute step test, in which patients had to lift their knee to a height midway between the patella and iliac crest, as much as possible within two minutes [34]. More steps during the two minutes indicate a higher physical capacity. Lower body strength was measured with the five time chair stand test, in which patients had to stand up five times from a chair as fast as possible, with less time indicating more strength [35]. Muscle strength was measured with a handheld dynamometer and was determined as the highest value of three attempts. A handgrip strength <27 kg for males and <16 kg for females indicates low strength. Furthermore, patients were asked to rate their, fatigue, fear of falling and pain on a ten-point numeric rating scale (NRS). Additionally, the presence of conditions as dyspnea, fatigue and syncope were asked. Other collected patient characteristics were age, sex, body mass index (BMI), living arrangement, surgical risk score (Euroscore-II), New-York heart association score (NYHA), chronic obstructive pulmonary disease (COPD), history of coronary heart disease, peripheral artery disease, living arrangement, left ventricular ejection fraction (LVEF) and access site of the procedure (transfemoral or transaortic).”

Geriatric depression score, median [IQR]  (table 1) points to a 5-point version? ( table 1).

The depression score in Table 1 has a score range between 0 and 15. We added GDS-15 to the table for clarification.